# FTIR Spectroscopy as a Tool to Study Age-Related Changes in Cardiac and Skeletal Muscle of Female C57BL/6J Mice

**DOI:** 10.3390/molecules26216410

**Published:** 2021-10-23

**Authors:** Sandra Magalhães, Idália Almeida, Filipa Martins, Fátima Camões, Ana R. Soares, Brian J. Goodfellow, Sandra Rebelo, Alexandra Nunes

**Affiliations:** 1iBiMED, Institute of Biomedicine, Department of Medical Sciences, University of Aveiro, Agra do Crasto, 3810-193 Aveiro, Portugal; sandra.vicencia@ua.pt (S.M.); idalia24@ua.pt (I.A.); samartins@ua.pt (F.M.); fatimacamoes@ua.pt (F.C.); ana.r.soares@ua.pt (A.R.S.); srebelo@ua.pt (S.R.); 2CICECO, Aveiro Institute of Materials, Department of Chemistry, Campus Universitário de Santiago, University of Aveiro, 3810-193 Aveiro, Portugal; brian.goodfellow@ua.pt

**Keywords:** aging fingerprint, aging muscle, spectroscopic profile

## Abstract

Studying aging is important to further understand the molecular mechanisms underlying this physiological process and, ideally, to identify a panel of aging biomarkers. Animals, in particular mice, are often used in aging studies, since they mimic important features of human aging, age quickly, and are easy to manipulate. The present work describes the use of Fourier Transform Infrared (FTIR) spectroscopy to identify an age-related spectroscopic profile of the cardiac and skeletal muscle tissues of C57BL/6J female mice. We acquired ATR-FTIR spectra of cardiac and skeletal muscle at four different ages: 6; 12; 17 and 24 months (10 samples at each age) and analyzed the data using multivariate statistical tools (PCA and PLS) and peak intensity analyses. The results suggest deep changes in protein secondary structure in 24-month-old mice compared to both tissues in 6-month-old mice. Oligomeric structures decreased with age in both tissues, while intermolecular β-sheet structures increased with aging in cardiac muscle but not in skeletal muscle. Despite FTIR spectroscopy being unable to identify the proteins responsible for these conformational changes, this study gives insights into the potential of FTIR to monitor the aging process and identify an age-specific spectroscopic signature.

## 1. Introduction

Aging studies are of great importance to elucidate this complex process, to detect and prevent age-related diseases, and to develop anti-aging therapies. However, the study of aging is still a work in progress for the scientific community, as there are several challenges that are difficult to overcome. A relevant obstacle to studying age is time; it is nearly impossible and totally impracticable to perform a longitudinal study of aging in humans; cost and ethical issues are other drawbacks that have to be considered when studying aging. Therefore, aging studies are mostly performed using different models, either computational models, cell cultures or animal models. Both in vitro and in vivo studies are widely used in aging research, using both cell and animal models [1,2]. Among all animal models used, *Mus musculus* mice are the preferred model for in vivo research for several reasons: they are small animals, which makes manipulation easier, have a relative short lifespan, a well-documented genome and they are similar to humans in many aspects of health and disease [1]. In addition, it is possible to correlate mice age to human age [3], making it easier to translate research results. However, despite all these advantages, mice do not appear to develop some of the most common age-related diseases, such as diabetes and atherosclerosis, so one needs to be careful when using this model to study aging and age-related diseases [3,4,5,6]. 

Studies on aging aim not only to predict the odds of the development of age-related diseases but also to identify the changes that occur during physiological aging and identify potential biomarkers of aging. This is a challenging task, and to date there are still difficulties in the identification of true biomarkers of healthy aging despite great efforts in different fields [7,8,9]. Whether using -omics analytic approaches (such as proteomics, lipidomics, transcriptomics, genomics or metabolomics) to identify molecules that specifically and significantly change throughout the aging process, or using clinical parameter such as blood pressure, bone density or maximal oxygen consumption, there are no perfect markers of physiological age yet, since they always also reflect changes that occur during pathogenesis [7]. One approach that has been gaining potential as a screening tool for the study of biological samples in health and disease is Fourier Transform Infrared (FITR) spectroscopy. This technique has already been used for the identification of cancer biomarkers [10], characterization of disease metabolites [11], identification of microorganisms [12], intra-operative identification of malignant tissue [13] and quantification of analytes in biological fluids [14], with robust and reproducible results. One of the great advantages of this approach is that it allows rapid screening of a given sample in order to obtain a spectroscopic profile characteristic of that sample with valuable information on the content of the main biological molecules: lipids, proteins, nucleic acids and carbohydrates. Although FTIR spectroscopy is not as specific as, for instance, NMR or mass-spectrometry approaches, it is less expensive and in the case of analysis of tissues, samples can be re-used for further analysis. In this way, it is a good starting point when initiating metabolomic analyses: one can understand the behavior of the sample, compare spectroscopic profiles in health and disease, and select which are the main groups of biomolecules that will be worth studying when using more invasive, time-consuming and expensive approaches.

Despite the entire organism suffering from the effects of aging, skeletal muscle is considered one of the most age-sensitive tissues, and is vital for several functions in the body, including respiration [15]. Additionally, cardiac muscle also undergoes deep changes during aging that can result in heart failure [16]. Thus, the aim of this study was to screen tissue samples of the cardiac and skeletal muscle of C57BL/6J female mice (40 mice samples) using FTIR spectroscopy and to assess the differences and similarities of changes in both tissues during aging. 

## 2. Results 

### 2.1. FTIR Spectra Overview and Pre-Treatments

To evaluate the spectroscopic profile of muscle tissue during aging, samples of skeletal and cardiac muscle of C57BL/6J female mice at 6, 12, 17 and 24 months of age were subjected to FTIR spectroscopy (ten biological and three technical replicates at each timepoint). 

The average baseline-corrected, area normalized spectra of skeletal and cardiac muscle are presented in Figure 1. Area normalization of the FTIR spectra was performed to ensure that differences in the amount of sample placed in the ATR crystal would not be the cause of spectral differences between samples. 

To perform a detailed analysis of age-related spectral changes, we analyzed each tissue independently, then compared both tissues to evaluate the main differences among them throughout the aging process. 

Raw spectra were subjected to PCA analysis for outlier detection and outlier removal. To further evaluate how age affects biomolecules, spectra were cut in three main spectral regions (3050–2800 cm^−1^, 1800–1500 cm^−1^ and 1200–900 cm^−1^), baseline-corrected, and area normalized and statistical analysis was performed. PLS-R analysis was performed to evaluate age-related changes in the tissue, and analysis of peak intensities was carried out to evaluate in detail some important peaks, namely those related to protein secondary structure. PLS analysis was performed individually in each spectral region for each tissue. For each region, the choice of which factor to use to interpret the results was performed in such a way as to maximize the variance explained by that factor and to avoid overfitting. In this way, for skeletal muscle, the best factors to use to discriminate between the samples were factor 1, factor 2 and factor 2 for the 3050–2800, 1800–1500 and 1200–900 cm^−1^ regions, respectively. For cardiac muscle, following the same logic, we used factor 3, factor 1 and factor 1 for 3050–2800, 1800–1500 and 1200–900 cm^−1^ regions, respectively (see Section 2.2 and Section 2.3 for detailed results).

### 2.2. Skeletal Muscle

As observed in Figure 1, the spectra of cardiac and skeletal muscle are visually identical, and one can only identify some slight differences in peak intensities when zoomed in, as in panel A of Appendix A. To identify in detail these spectral differences, we further analyzed the three main spectroscopic regions, 3050–2800 cm^−1^; 1800–1500 cm^−1^ and 1200–900 cm^−1^ with PLS analysis. Briefly, we took second-derivative spectra of each normalized spectral region and performed a PLS analysis using the age of the mice as the Y matrix and spectral data as the X matrix. As pointed out at the end of Section 2.1, for each spectral region we chose the number of factors that allowed for the best discrimination between samples without overfitting. Appendix A shows the second-derivative spectra of skeletal muscle tissue in the 3050–2800 cm^−1^ region, used for PLS model. The PLS score plot is shown in Appendix A, and corresponding loadings are shown in Appendix A. Factor 1 partially discriminates younger animals (6 M) from older animals (24 M), with a sensitivity of 80% and a specificity of 75%. The factor 1 loadings (Appendix A blue line) suggest that the peak at 2925 cm^−1^, assigned to CH_2_ groups from lipid acyl chains, is associated with older samples and the peak at 2868 cm^−1^, assigned to CH_3_ groups of lipid acyl chains, is related to younger samples. 

To further evaluate if there are changes in the intensity of some specific peaks during aging, we performed a peak intensity analysis using second derivative spectra and normalized spectra, as described in the methods section. Specifically, in the 3050–2800 cm^−1^ region, we evaluated the acyl chain length and lipid unsaturation levels. The results showed no significant changes during aging (Figure 2A,B black bars); however, a tendency for an increase in lipid unsaturation from 12 M to 24 M can be observed (Figure 2B black bars). 

The 1800–1500 cm^−1^ spectral region is especially important to analyze changes in protein secondary structure using the Amide I and Amide II peaks. Appendix A shows the second derivative spectra of skeletal muscle. PLS results show that there is a positive correlation between the spectral profile in this region and the age of the mice (correlation coefficient R = 0.72). Moreover, factor 2 discriminates samples of younger mice from those of older mice with a sensitivity of 86.7% and a specificity of 87.5% (Appendix A). The analysis of the loadings plot (Appendix A) shows that peaks at 1741 cm^−1^ assigned to carbonyl groups, at 1651 cm^−1^ assigned to α-helix structures of proteins, and at 1540 and 1512 cm^−1^, both assigned to Amide II of proteins, are associated with older samples (24 M), while peaks at 1693 cm^−1^ assigned to antiparallel β-sheets, at 1662 cm^−1^ assigned to β-sheets, at 1625 cm^−1^ assigned to intermolecular β-sheets, and at 1554 cm^−1^ assigned to Amide II of proteins, are associated with samples from six month old mice. 

The analysis of peak intensities in this region revealed that the levels of triglycerides (TG) as estimated by the intensity of the peak at 1741 cm^−1^ and assigned to carbonyl groups (C=O), do not vary significantly during aging (Figure 2C black bars), a behavior also seen for total protein levels (Figure 2D black bars). Figure 2E–G (black bars) shows there are no differences in the intensities of peaks related to the secondary structure of proteins. 

In the so-called *fingerprint region* (1200–900 cm^−1^), it is possible to observe bands that arise mainly from carbohydrates, nucleic acids and lipids. The second derivative spectra of skeletal muscle are presented in Appendix A. Similar to the 1800–1500 cm^−1^ region, PLS-R analysis showed a positive correlation between the spectral profile and the age of the tissue (correlation coefficient R = 0.81). The score plot (factor 2 vs. factor 3) discriminates samples of six month old mice from the oldest samples (24 months) with a sensitivity of 90% and a specificity of 75% (Appendix A). The corresponding loadings show that a peak at 1045 cm^−1^, which may arise from glucose, is associated with samples from 24 month old mice, while peaks at 1155 and 1081 cm^−1^, assigned to glycogen and PO_4_^-^ groups of DNA, respectively, are related to samples from six month old mice (Appendix A). Despite the PLS analysis showing that glucose may be related to older samples, the analysis of the peak intensities revealed no significant changes during aging (Figure 2I black bars). The peak intensity analysis of the peak related to cholesterol esters (1169 cm^−1^) also showed no variation with aging (Figure 2H black bars). 

### 2.3. Cardiac Muscle

As can be seen in Figure 1B, there is a decrease in the intensity of the Amide I and II peaks as well as in peaks assigned to CH_2_ and CH_3_ groups in the cardiac muscle of older mice when compared with younger animals; thus, these differences were evaluated using PLS and peak intensity analysis. To see in detail the differences in peak intensities between younger and older animals, the three main spectral regions were zoomed in and plotted in panel A of Appendix A. For PLS analysis we used second-derivative spectra of each normalized spectral region and used the age of the mice as the Y matrix and spectral data as the X matrix. As pointed out at the end of Section 2.1, we chose the number of factors that allowed for the best discrimination between samples without overfitting for each spectral region.

The average second derivative spectra of mice cardiac muscle of all age groups in the 3050–2800 cm^−1^ region are presented in Appendix A. A PLS model was built using these spectra, and a PLS score plot is presented in Appendix A. Factor 3 discriminates samples of six month old mice (negative factor 3) from older samples, which are located mainly in the positive sector of factor 3, with a sensibility of 63% and a specificity of 87.5%. The peaks responsible for this discrimination are highlighted in Appendix A in the loadings plot. The results clearly show that peaks at 3013 and 2877 cm^−1^, assigned to the olefinic band (CH group of double bands) and CH_3_ groups, respectively, are associated with older samples, and the peak at 2851 cm^−1^, from CH_2_ groups, is associated with six month old mice. The intensity of some peaks was also analyzed in order to calculate the length of acyl chains and unsaturation levels (Figure 2A,B grey bars). The results show no significant differences throughout aging in any of the calculated levels.

The Appendix A presents the second derivative spectra of cardiac muscle in the 1800–1500 cm^−1^. PLS analysis revealed a positive correlation between age and spectral profile in this region, with a correlation coefficient R = 0.70. Looking to the score plot, there is a clear discrimination in factor 1 of the younger samples (6 months) from older samples (24 months) (Appendix A), with a sensitivity of 66.7% and a specificity of 87.5%. The loadings (Appendix A) show that peaks at 1744 (carbonyl groups), 1682 (β-sheet structures of proteins), 1648 (α-helix structures of proteins), 1625 (intermolecular β-sheet structures of proteins) and 1554, 1540 and 1512 cm^−1^ (Amide II of proteins) are associated with older samples (24 M), and peaks at 1696 (antiparallel β-sheets of proteins) and 1662 cm^−1^ (β-turns of proteins) are associated with younger samples (6 M).

The analysis of peak intensities showed no significant differences in the levels of triglycerides, total protein levels, antiparallel β-sheets, intermolecular β-sheets and fibril formation during aging (Figure 2C–G grey bars).

The second derivative spectra of cardiac muscle in the fingerprint region (1200–900 cm^−1^) are presented in Appendix A. PLS analysis showed that there is a positive strong correlation between the spectroscopic profile and the age of the tissue, with a correlation coefficient R = 0.78.

Looking to the score plot, a clear distinction is noticeable between the samples of six month old mice and samples from 24 month old mice by factor 1, with a sensibility of 92.6% and a specificity of 87.5% (Appendix A). The loadings plot (Appendix A) shows that this discrimination is explained by peaks related to cholesterol esters (1166 cm^−1^) and glucose (1050 cm^−1^), which appear to be related to older samples; however, analysis of peak intensities of these two peaks revealed no significant changes during the aging process (Figure 2H,I grey bars).

### 2.4. Skeletal vs. Cardiac Muscle

To compare the behavior of both striated muscle tissues during aging and highlight the major differences between these two tissues, we performed a comparative analysis of all spectra of skeletal and cardiac muscle in the three spectral regions by PCA. In the 3050–2800 cm^−1^ region, PCA separated skeletal and cardiac muscle by PC1, with a sensitivity of 82.4% and a specificity of 83.8% (Appendix A). In the 1800–1500 cm^−1^ region, the differences between tissues are even more evident, with a sensitivity of 100% and 99.1% specificity by PC1 separation (Appendix A). In the fingerprint region, PC1 also discriminates cardiac from skeletal muscle, with a sensitivity of 91.9% and a specificity of 98.2% (Appendix A). 

The analysis of peak intensities revealed significant differences between the two tissues. Concerning lipids, there is a tendency for acyl chains to be bigger in cardiac muscle, although this difference is only significant for 24 months (Figure 2A). Similarly, lipid unsaturation levels are higher in cardiac muscle, being significant different in all ages (Figure 2B). As concerns triglycerides, there are no significant differences in the levels of these compounds between skeletal and cardiac muscle (Figure 2C). Analysis of peak intensities also showed that cardiac muscle has lower levels of proteins than skeletal muscle; however, this difference was only significant for older mice (17 and 24 months) (Figure 2D). With respect to differences in protein secondary structure between the two tissues, the results show no significant differences between both tissues in the amount of antiparallel β-sheets (Figure 2E); however, lower levels of intermolecular β-sheets were observed in cardiac muscle compared to skeletal muscle for all analyzed ages (Figure 2F). Regarding cholesterol esters, cardiac muscle seems to have higher levels than skeletal muscle, but this difference is not statistically significant (Figure 2H). On the contrary, the results indicate that cardiac muscle has lower levels of glucose than skeletal muscle; however, this result was also not significant (Figure 2I).

## 3. Discussion

The present study attempted to evaluate age-related changes in the spectroscopic profiles of cardiac and skeletal muscle from mice and to identify spectroscopic markers of aging that would allow us to classify the status of the tissues in terms of age.

Mice are the preferred models for human aging studies, as it is possible to roughly translate the age of mice into human age [3]. We used C57BL/6J female mice at 6, 12, 17 and 24 months of age, which translate to mature adults (6 M), middle aged (12 M) and old aged mice (17 and 24 M) [6]. The equivalent age range in humans would therefore be from around 30 years of age to around 70, which, taking into account the average lifespan, can allow for some understanding of the metabolic changes concomitant with this physiological process. However, the fact that we used only female mice for this study prevents us from carrying out an accurate approximation of the human physiological response, as animals from both sexes would be needed to predict the response in humans in the most accurate way [17]. In addition, changes in the body composition of mice that are inherent to their aging can bring additional bias into these types of studies.

Both skeletal and cardiac muscle are widely affected by aging. Sarcopenia, for instance, is a common feature of aging and impairs skeletal and cardiac muscle function, decreasing autonomy [18,19,20]. Even without comorbidities such as cardiovascular disease, an association between skeletal and cardiac muscle sarcopenia has been found, reinforcing the link between these two muscle types during aging [21]. There is also evidence that aging increases acetylation of key proteins in both striated muscles, which may also contribute to muscle deterioration [22]. Given that the global population is aging, understanding the molecular mechanisms underlying muscle aging can allow for the development of therapeutic strategies to improve quality of life for the elderly. 

The FTIR results for the 3000–2800 cm^−1^ spectral region seem to indicate that the skeletal muscle of older animals has longer lipid chains than younger mice. On the other hand, in cardiac muscle it seems that there is a tendency for a decrease in CH_2_ groups and an increase in CH_3_ and CH groups in lipids upon aging, which may indicate that older cardiac muscle lipids have shorter carbon chains. In addition, the increase in the amount of CH groups may indicate an increase in unsaturation levels in this tissue with age (although the analysis of peak areas did not reveal any significant differences in unsaturation levels in this tissue upon aging). Houtkooper et al. studied metabolic fingerprints from the plasma, liver and muscle of aging C57BL/6J mice [22]. Results showed an increase in the levels of free fatty acids and a decrease in long-chain acylcarnitines in the plasma of 24 month old mice. They also reported an increase in saturated 18:0 and 20:0 free fatty acids and a decrease in 16:0 and 16:2 FFAs in older mice [22]. More importantly, these authors identified decreased levels of linoleic acid from erythrocytes and decreased levels of C16 acylcarnitines as strong predictors of aging. In muscle, the authors detected 65 metabolites that changed with age, most of them related to fatty acid metabolism [22]. These results indicated that lipids suffer alterations throughout the aging process and may be used to predict age [23]. In another study, Zhou et al. found an increase in 18:0 ceramides in aged skeletal muscle [24], and Wong et al. [25] studied the lipidome of human plasma during aging and found a generalized decrease in the levels of all lipid classes in older individuals, independently of sex and body mass index.

The FTIR results for the 1800–1500 cm^−1^ region provide insights into changes in the protein secondary structure in skeletal and cardiac muscle during aging. Specifically, in skeletal muscle, our data from PLS analysis indicate a decrease in both antiparallel and intermolecular β-sheets in β-sheet-containing proteins upon aging. This could be due to a decrease in the expression of β-sheet rich proteins or due to a modification in the secondary structural elements of existing proteins towards structures with less β-sheet. In cardiac muscle, the results show an increase in the content of intermolecular β-sheets and a decrease in antiparallel β-sheets. It is widely known that aging causes a progressive decline in proteostasis and a consequent increase in protein aggregation levels in several organisms (known as metastable proteins) [26,27]. This is often associated with increased levels of β-sheet structures, since this secondary structure is aggregation-prone and is found in proteins present in aggregates of known neurodegenerative diseases. Tanase et al. reported an increase in protein aggregation levels in the bone marrow and spleen of 22 month old mice compared to three month old mice [28]. Leeman et al. also reported increasing protein aggregation levels in the neural stem cells of aged mice, due to defects and reduced activity of the lysosomal pathway [29]. It has also been reported that deposition of amyloid proteins occurs in the cardiac muscle of mice and that cardiac amyloidosis is not as rare as it was thought to be [30,31]. Therefore, our results from skeletal muscle seem to contradict results reported from other cells and tissues, where a tendency for a decrease in β-sheets during aging in this tissue is seen. However, this may not mean a decrease in total protein aggregation levels. In fact, FTIR spectroscopy only detects changes in protein secondary structure, and in this case, we observed a decrease in β-sheets concomitant with age. Nevertheless, it is known that only a small fraction of proteins that aggregate during the aging process have aggregation-prone structures, such as poly-Q chains or increased β-sheets [27,32]. Therefore, the decrease in β-sheet structures observed here may not reflect a decrease in the amount of protein aggregates, but rather a decrease in the expression of proteins with this structure. Future work using complementary approaches such as SDS-PAGE and mass spectrometry will be performed to help elucidate the changes in protein aggregation pattern with age and identify specific proteins that are aggregating. In cardiac muscle, the increase in intermolecular β-sheets with age may indicate that proteins aggregating in this tissue are different from those aggregating in skeletal muscle. It would be of great interest to perform a comparative study between these two tissues in order to evaluate differences, for instance, in the proteomic protein aggregation profile with age. In addition, despite the levels of intermolecular β-sheets appearing to decrease with age, a comparison between cardiac and skeletal muscle indicates that skeletal muscle has higher levels of these structures for all ages analyzed. Finally, carbonyl groups (peak at 1741 cm^−1^) in this region show an increase in the levels of C=O bonds with age for both tissues, which may indicate, for instance, higher phospholipid content, something that has already been reported for aged skeletal muscle [33]. 

FTIR analysis of the fingerprint region (1200–900 cm^−1^) showed an increase in glucose levels in older mice in both tissues, and increased glycogen levels in the skeletal muscle of younger mice. These results are in agreement with the literature, where Uchitomi et al. found decreased levels of compounds associated with glucose metabolism, in particular fructose 1,6-diphosphate, in the skeletal muscle of aged mice (28 months-old) that they associated with a decrease in the glycolytic pathway with age [33]. Also, Houtkooper et al. [34] found increased levels of glucose in liver and skeletal muscle of aged mice.

One of the limitations of this study is that FTIR spectroscopy is not able to identify specific molecules, only functional groups of molecules. This prevents us from reaching definitive conclusions about the results; we can only infer potential changes in different groups of biomolecules that are characterized mainly by a given functional group in a given region of the spectra (e.g., despite there being several molecules with C=C and C–H bonds, the spectral region between 3050–2800 cm^−1^ is characteristic of lipids). Other limitations of the study include the fact that we only analyzed total protein, without any fractioning. Thus, changes detected in protein conformation in the FTIR spectra cannot be attributed to specific proteins but must rather be analyzed as a whole. Further studies using proteomic approaches are needed, and some are already ongoing to identify specific proteins which undergo conformation changes during the physiological aging process.

This work highlights the potential of FTIR spectroscopy as a tool to identify age-related changes in mouse muscle. The results suggest differing changes during aging for cardiac and skeletal muscle, indicating that these tissues age differently, given they present a different protein secondary structure spectroscopic signature. The results also revealed that there is a decrease in antiparallel β-sheets with age, characteristic of oligomeric structures in both cardiac and skeletal muscle. However, while there is an increase in aggregation-prone intermolecular β-sheet structures in proteins of cardiac muscle, there is not in skeletal muscle. Since human samples of cardiac muscle are not easily accessible, the study of murine samples may open doors to understanding the potential impact of increase in aggregation-prone structures in this tissue and evaluating the potential impact it may have on human health. In this way, targeted therapies may be considered to slow the deteriorative effects of aging on the heart. Despite FTIR spectroscopy not being able to identify specific molecules, it is a cheap, reliable and accessible tool to monitor spectral signatures specific to tissue aging.

## 4. Materials and Methods

### 4.1. Animals

Colonies of female C57BL/6J mice were obtained from Charles River Laboratories, UK, at 6, 12, 17 and 24 months of age, and allowed to acclimatize in a vivarium for at least a week before being euthanized. Animals were maintained under a controlled environment (23 °C and 12 h light cycle with food and water ad libitum), and their health status and well-being were monitored daily by the iBiMED animal facility staff. The animal study protocol was reviewed and approved by the Medical Sciences Department of the University of Aveiro animal welfare body (approval number 01/2018). Briefly, the animals were euthanized by cervical dislocation followed by decapitation. An incision was made in the thorax to collect the heart (from now on designated cardiac muscle, its major component). In addition, a portion of skeletal muscle was collected from the limbs after removing the skin (from now on designated skeletal muscle, its major component). The heart and the skeletal muscle were immediately frozen in liquid nitrogen and subsequently transferred to −80 °C. 

### 4.2. Tissue Preparation

From a total of 40 animal samples comprising four groups, 10 young—6 months (6 M), 10 adult—12 months (12 M), 10 middle-aged—17 months (17 M) and 10 old—24 months (24 M), the heart and skeletal muscle were collected, immediately frozen in liquid nitrogen and stored at −80 °C. For analysis, frozen samples were pulverized in a dry ice cooled mortar and a 10 mg specimen used for FTIR measurements.

### 4.3. FTIR Measurements

All FTIR spectra were acquired in ATR mode with an FTIR Bruker Alpha Platinum spectrometer (Bruker^©^, Billerica, MA, USA) coupled to OPUS software (Bruker^©^, Billerica, MA, USA). Spectra were obtained in the mid-infrared range (4000–600 cm^−1^) with a resolution of 8 cm^−1^ and 64 co-added scans. Spectral acquisition was performed in a room with controlled conditions of temperature and relative humidity (23 °C and 35%, respectively). Tissue samples were placed at the center of an ATR diamond crystal with a spatula and left to air dry. The drying process was accompanied by visual observation of the live spectrum with OPUS software; the sample was considered dried when spectrum profile did not change. For each age and for each tissue type, 10 biological replicates and 3 technical replicates were acquired. A background spectrum was acquired against air between each sample, and the ATR crystal was cleaned with 70% ethanol and distilled water.

### 4.4. FTIR Data Processing and Analysis

#### Preprocessing

All spectra were exported in OPUS format and imported to The Unscrambler X software (V.10.5., Camo Analytics). All spectra were individually visually analyzed and spectra with background noise or with suspicious profile were repeated to ensure good quality data. After visual inspection, we separately performed a PCA analysis on both skeletal and cardiac muscle spectra in order to identify and remove outliers. Samples with high values for PCA Q-residuals were considered outliers and removed from the data matrix. 

After outlier removal, spectra were divided into three spectral regions: 3050–2800 cm^−1^, 1800–1500 cm^−1^ and 1200–900 cm^−1^. Sub-spectra were then baseline corrected and area normalized. Normalization was performed in order to ensure differences in the amount of sample placed in the crystal were not responsible for differences in the spectral profiles. Normalized spectra were then derived using the 2nd derivative with Savitzky–Golay algorithm and three smoothing points. Since spectra from biological samples are complex, with several overlapping peaks, the use of the derivative is crucial to resolve the peaks and extract any valuable information. The pre-processed spectra were then subjected to both multivariate analysis (PCA and PLS) and analysis of specific peak intensity.

### 4.5. Multivariate Analysis: PCA and PLS

Spectral data has thousands of variables (spectral points) that would be impossible to analyze individually. Multivariate analysis allows for the reduction spectral data to fewer variables, called principal components (PCs) in PCA analysis and factors in PLS analysis. In each analysis, for each dataset, one must choose the best PCs or factors to use in order to explain the results in a way that allows extraction of the most valuable biological information without overfitting (see [35] for detailed information).

To analyze changes in the spectral profiles of both cardiac and skeletal muscle during aging, we performed a PLS analysis on both tissues individually in the three above-mentioned spectral regions. The PLS model was built using the 2nd derivative spectra and a random intern cross-validation and Kernel algorithm. PLS analysis produces a scores plot, which is a scatter plot with a projection of the data in two dimensions. Since PLS is a supervised multivariate statistical test, one has two matrices of data (X and y), in this case, the spectral data and the age of the mice, respectively. Besides the score plot, PLS analysis produces a loadings plot that explains discrimination.

To compare cardiac and skeletal muscle, we performed a PCA analysis on all three spectral regions, using the 2nd derivative spectra and up to seven principal components. All multivariate analyses were performed using The Unscrambler X software (v.10.5 CAMO Analytics).

### 4.6. Intensity of Spectral Bands

To calculate the intensity of the spectral bands we used different approaches: for the calculation of intensity of peaks assigned to CH (3013 cm^−1^), CH_2_ (2851 cm^−1^ and 2922 cm^−1^) CH_3_ (2959 cm^−1^ and 2871 cm^−1^), C=O (1741 cm^−1^), glucose (1045 cm^−1^), cholesterol esters (1169 cm^−1^) and protein secondary structures, namely β-sheets (1693 cm^−1^, 1682 cm^−1^ and 1628 cm^−1^), we inverted 2nd derivative correspondent spectra by factoring by −1, as previously described [36,37]. Then we selected the wavenumbers corresponding to that peak and extracted the intensity values. The use of 2nd derivative spectra for these calculations was due to the need to resolve overlapping signals and ensure correct information.

For calculation of the fibril formation ratio and total protein amount we used non-derivative baseline corrected and normalized spectra to extract the values of the intensity of the Amide I and Amide II peaks. 

Statistical analysis was performed together for both tissues with GraphPad Prism 6 software (GraphPad Software, Inc.), using Ordinary Two-Way ANOVA (not repeated measures) and the Sidak test for multiple comparisons of all means, with a confidence level of 0.05. 

## Figures and Tables

**Figure 1 molecules-26-06410-f001:**
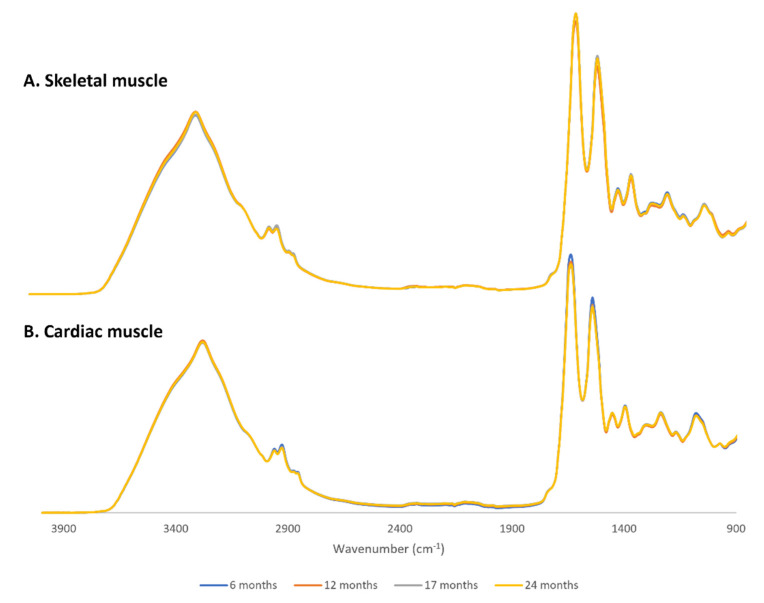
Baseline-corrected, area normalized average FTIR spectra of skeletal (**A**) and cardiac (**B**) muscle in the mid-infrared range (4000–900 cm^−1^).

**Figure 2 molecules-26-06410-f002:**
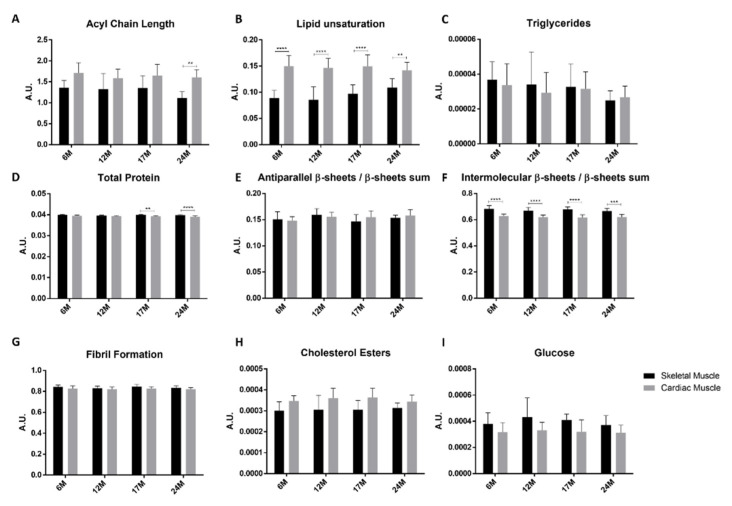
Analysis of peak intensities of cardiac and skeletal muscle spectra. (**A**) Acyl chain length, calculated using CH_2_ and CH_3_ peak intensities (ratio I_2851+2922_/I_2959+2871_); (**B**) Lipid unsaturation levels, calculated using the ratio between olefinic band and CH_2_ bands (ratio I_3013_/ I_2851+2922_); (**C**) The levels of triglycerides, calculated using intensity of C=O band at 1741 cm^−1^; (**D**) Total protein levels, calculated by the sum of Amide I and Amide II peaks (I_Amide II_/I_Amide I_); (**E**) Ratio of antiparallel β-sheet/β-sheets sums, calculated using I_1693_/I_1693+1682+1628_; (**F**) Ratio of intermolecular β-sheets/β-sheets sums, calculated using I_1628_/I_1693+1682+1628_; (**G**) Fibril formation, calculated using Amide I and Amide II peak intensities (ratio I_Amide II_/I_Amide I_) using baseline-corrected, area-normalized and non-derived spectra; (**H**) Cholesterol ester levels, calculated using the intensity of peak at 1169 cm^−1^; and (**I**) Glucose levels, calculated using the intensity of peak at 1045 cm^−1^. Data are expressed as mean ± SD. ** *p* < 0.01; *** *p* < 0.001; **** *p* < 0.0001.

## Data Availability

Not applicable.

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
