# Peer review of "FTIR Spectroscopy as a Tool to Study Age-Related Changes in Cardiac and Skeletal Muscle of Female C57BL/6J Mice"

_molecules, 2021, doi:10.3390/molecules26216410_

Round 1

Reviewer 1 Report

The aim of this study was to examine cardiac and skeletal muscle tissue samples in female C57BL / 6J mice using FTIR spectroscopy and to evaluate differences and similarities of these changes in both tissues through aging. Surely, to validate the obtained results, the authors must consider additional complementary techniques, so to clarify the changes in the protein aggregation pattern that occur with advancing age and identify specific proteins that are aggregating. Furthermore, among their deductions, the authors suggest that carbonyl groups (peak at 1741 cm-1) show increased C = O bond levels with age for both tissues and point to a phospholipid content as a possible explanation. I would suggest that the authors also investigate the oxidative processes typical of aging in a future paper. 

Author Response

Dear reviewer, thank you for your suggestions. We completely agree with your comments. In fact, this is the first experimental approach on these samples. At the moment we are starting a proteomic study on these samples that we want to publish as soon as we have the results, to consolidate this study. We know FTIR spectroscopy is not a specific approach, that it does not allow to identify specific molecules, but we consider this a promising exploratory study to open doors to more detailed and expensive methods. Concerning your suggestion about oxidative processes we thank you for your contribute and it is in fact very pertinent to look at oxidative response in our future work with this samples.

Reviewer 2 Report

This study gives insights into the potential of FTIR to monitor the aging process and identify an age-specific spectroscopic signature. The design of this study was good, results from this submission are sound, however, there are still some issues that need to be revised. A revision is suggested.

  1. Line 11, please delete “1” before the Abstract.
  2. Please define FTIR in the abstract.
  3. Please provide the detail information of cardiac and skeletal muscle isolation, which areas were collected from animals.

4.Please provide the detail information of animal study, including the way for sacrifice, anaesthetization …

  1. Please discuss the limitation of this study.
  2. Please strength the clinical implications of this study.

Author Response

We thank reviewer 2 for the suggestions. Here are our answers.

  1. Line 11, please delete “1” before the Abstract. Changes were made in the document.
  2. Please define FTIR in the abstract. Changes were made in the document.
  3. Please provide the detail information of cardiac and skeletal muscle isolation, which areas were collected from animals. We add this information in the Materials and Methods section.
  4. Please provide the detail information of animal study, including the way for sacrifice, anaesthetization … We add this information in the Materials and Methods section.
  1. Please discuss the limitation of this study. We add a paragraph about the limitations in the discussion section (lines 315-25).
  2. Please strength the clinical implications of this study. We add two sentences about the clinical implications at the end of the discussion section (lines 333-337).

Reviewer 3 Report

This manuscript aims to use FTIR spectral analysis on skeletal and cardiac tissue in mice to correlate spectral features with aging. The concept and execution of the study is good, but there are issues with the explanation of the results. The problem is not that the results are bad, indeed I think things are sound, but explanation of PCA factors, the meaning behind the intermediate spectral findings, and just exactly what has been done between taking the spectra and producing the resulting conclusions is at times confusing, unclear, or absent. I believe once the level of clarity is improved in this data analysis/results this paper will be acceptable.

My detailed comments and suggestions are as follows.
Ln 31 remove "very"
ln 44 this sentence reads as if using a cross-section of aged mice overcomes the issue of lack of commonage-related disease, but I do not this this solves this. It is unclear. Please clarify/revise.

ln 100. I disagree somewhat. Maybe visually difficult to distinguish differences in panel A, but in the ~ 2900, 1700, 1500, 1000 peak areas the intensities clearly seem to decrease with age in panel B. I suggest zooming in to the 3 target spectral ranges and plotting these for the reader. Perhaps it can be in SI, where we can only see detail in the derivative spectra. Perhaps one region (pick what you think is most important or most change) could be zoomed in and inset in the panels A and B in the largely empty region above the trace from ~2200-1800 cm-1.

Either in ~ on 94-98 or early in sec. 2.2 you need a sentence or two explaining how the changes in your 3 regions are related to Factor 1, Factor 2, etc. This is not explained and is confusing. Please try to clarify.
Continuing on with this same thought, line 102 says "... were analyzed... as follows" but then we just are pointed to results in another document, not explained the analysis. At this point, please add a sort of summary sentences that explain what you did with the analysis so that what follows makes more sense to the reader. Something on the level of "we took derivative spectra, the changes in these spectra were quantified, and analyzed with PLS?PCA? this resulted in 3 correlation Factors.." Or however it goes. I understand the basics of this, but it is still not clear to me about the Factors.
Moreover, the actual nature of Factor 1,2,3 are not clear. Are these the principle components from PCA? Do they mean the same thing across all three regions (across all SX figures?) or are they unique to each set of SX figures/analyses? This is never defined, so we don't really know what these factors refer to. This must be explained better.

ln 159-162 I think it makes more sense to refer to this as a decrease in older mice, relative to younger. Best to keep the logic of changes in aging characteristics chronological with age.

ln 193-194 "Scores Plot" "Factor 1" sometimes these are capital letters, sometimes lowercase in the text. Pick one and be consistent throughout.
Also, the Scores Plot I don't feel is adequately explained anywhere.

ln 204 is PC1 the same as Factor 1??? Again these things just appear in the text with no explanation of what they are.

ln 213 starting "In what concerns triglycerides..." The sentence is awkward and grammatically weird, in a way I can't quite figure out. please revise.

ln 228 remove "one" I think.

ln 342- The Bruker Alpha should have a spectral resolution as small as 2 cm-1. Why use 8 cm-1 resolution? Especially when have a large signal in only 64 scans, and you are looking for small feature changes. Were any experiments run with smaller resolution? I wonder if any fine structure is lost at 8 cm-1. It seems that you would want the best resolution possible to distinguish changes in features. Please comment on this. My guess is that more/better information could be extracted from the spectral analysis with 2cm-1 resolution, even if at this resolution, more scans would be needed for similar signal to noise. With that instrument, the difference is likely a 5 minute scan instead of a 1 min scan, so that is not an unreasonable trade off time-wise. Seems like a missed opportunity for better data.

Author Response

We thank reviewer 3 for the suggestions. Below you can find our answers.

Ln 31 remove "very". Changes were made in the document.

ln 44 this sentence reads as if using a cross-section of aged mice overcomes the issue of lack of common age-related disease, but I do not this this solves this. It is unclear. Please clarify/revise.

We thank reviewer 3 for this comment. We reviewed the sentences, and it was, in fact, confuse. We remove the last part of the sentence. We think it is clearer now.

ln 100. I disagree somewhat. Maybe visually difficult to distinguish differences in panel A, but in the ~ 2900, 1700, 1500, 1000 peak areas the intensities clearly seem to decrease with age in panel B. I suggest zooming in to the 3 target spectral ranges and plotting these for the reader. Perhaps it can be in SI, where we can only see detail in the derivative spectra. Perhaps one region (pick what you think is most important or most change) could be zoomed in and inset in the panels A and B in the largely empty region above the trace from ~2200-800 cm-1.

We thank reviewer 3 for this comment. It is a very good observation. We zoomed in the three main spectral regions of both skeletal and cardiac muscle and insert it in supplementary figures 1-6, respectively. We also add that information on sections 2.2 and 2.3.  

Either in ~ on 94-98 or early in sec. 2.2 you need a sentence or two explaining how the changes in your 3 regions are related to Factor 1, Factor 2, etc. This is not explained and is confusing. Please try to clarify.

We thank reviewer 3 for this comment. We add a paragraph at the end of section 2.1 trying to explain this.

Continuing with this same thought, line 102 says "... were analyzed... as follows" but then we just are pointed to results in another document, not explained the analysis. At this point, please add a sort of summary sentences that explain what you did with the analysis so that what follows makes more sense to the reader. Something on the level of "we took derivative spectra, the changes in these spectra were quantified, and analyzed with PLS? PCA? this resulted in 3 correlation Factors.." Or however it goes. I understand the basics of this, but it is still not clear to me about the Factors.

Thank you for the suggestion. We add a short paragraph at the beginning of sections 2.2 and 2.3 to clarify the analysis.

Moreover, the actual nature of Factor 1,2,3 is not clear. Are these the principal components from PCA? Do they mean the same thing across all three regions (across all SX figures?) or are they unique to each set of SX figures/analyses? This is never defined, so we don't really know what these factors refer to. This must be explained better.

We thank reviewer 3 for this comment. We think that the text we added at the end of section 2.1 helps to clarify this.

ln 159-62 I think it makes more sense to refer to this as a decrease in older mice, relative to younger. Best to keep the logic of changes in aging characteristics chronological with age.

Thank you for the suggestion. Changes were made in the document.

ln 193-94 "Scores Plot" "Factor 1" sometimes these are capital letters, sometimes lowercase in the text. Pick one and be consistent throughout.

Thank you for the suggestion. Changes were made in the document.

Also, the Scores Plot I don't feel is adequately explained anywhere.

We thank reviewer 3 for this comment. We add an explanation about scores plot in the section 4.5, that we think helps to explain the graphs.

ln 204 is PC1 the same as Factor 1??? Again these things just appear in the text with no explanation of what they are.

We thank reviewer 3 for this comment. We add an explanation about this in the section 4.5, that we think helps to better explain these features, but briefly PCs refer to PCA analysis and factors refer to PLS analysis.

ln 213 starting "In what concerns triglycerides..." The sentence is awkward and grammatically weird, in a way I can't quite figure out. please revise.

We thank reviewer 3 for the suggestion. We revise the sentence, and we think it is clearer now.

ln 228 remove "one" I think.

Thank you for the suggestion. Changes were made in the document.

ln 342- The Bruker Alpha should have a spectral resolution as small as 2 cm-1. Why use 8 cm-1 resolution? Especially when have a large signal in only 64 scans, and you are looking for small feature changes. Were any experiments run with smaller resolution? I wonder if any fine structure is lost at 8 cm-1. It seems that you would want the best resolution possible to distinguish changes in features. Please comment on this. My guess is that more/better information could be extracted from the spectral analysis with 2cm-1resolution, even if at this resolution, more scans would be needed for similar signal to noise. With that instrument, the difference is likely a 5-minute scan instead of a 1 min scan, so that is not an unreasonable trade off timewise. Seems like a missed opportunity for better data.

We thank reviewer 3 for the suggestion. It is absolutely right that with a lower resolution it would be possible to extract more information. In fact, in the beginning of our experiments, when we were optimizing the acquisition conditions, we tried to use 2cm-1 resolution on these samples, however, unfortunately the SNR was compromised. The best conditions we achieved were with a 8cm-1 resolution. The time of analysis was not a even a question that was raised as, as you said, it is a 4-5min scan. Despite this, one thing that made us more comfortable with our choice was that this resolution is used in several studies of spectroscopy in biological samples, as you can see in our recent review paper (Magalhães, S. et al. FTIR spectroscopy in biomedical research: how to get the most out of its potential, Applied Spectroscopy Reviews, 2021, doi:  10.1080/05704928.2021.1946822).

Round 2

Reviewer 2 Report

My questions had been well addressed, this paper is acceptable.